# Melanin’s Journey from Melanocytes to Keratinocytes: Uncovering the Molecular Mechanisms of Melanin Transfer and Processing

**DOI:** 10.3390/ijms241411289

**Published:** 2023-07-10

**Authors:** Liliana Bento-Lopes, Luís C. Cabaço, João Charneca, Matilde V. Neto, Miguel C. Seabra, Duarte C. Barral

**Affiliations:** iNOVA4Health, NOVA Medical School, Faculdade de Ciências Médicas, NMS, FCM, Universidade NOVA de Lisboa, 1169-056 Lisboa, Portugal; liliana.lopes@biol.ethz.ch (L.B.-L.); luis.cabaco@nms.unl.pt (L.C.C.); joao.charneca@nms.unl.pt (J.C.); matilde.neto@nms.unl.pt (M.V.N.); miguel.seabra@nms.unl.pt (M.C.S.)

**Keywords:** melanin, melanocyte, keratinocyte, melanosome, melanocore, melanokerasome, melanin transfer, melanin processing, melanin polarization

## Abstract

Skin pigmentation ensures efficient photoprotection and relies on the pigment melanin, which is produced by epidermal melanocytes and transferred to surrounding keratinocytes. While the molecular mechanisms of melanin synthesis and transport in melanocytes are now well characterized, much less is known about melanin transfer and processing within keratinocytes. Over the past few decades, distinct models have been proposed to explain how melanin transfer occurs at the cellular and molecular levels. However, this remains a debated topic, as up to four different models have been proposed, with evidence presented supporting each. Here, we review the current knowledge on the regulation of melanin exocytosis, internalization, processing, and polarization. Regarding the different transfer models, we discuss how these might co-exist to regulate skin pigmentation under different conditions, i.e., constitutive and facultative skin pigmentation or physiological and pathological conditions. Moreover, we discuss recent evidence that sheds light on the regulation of melanin exocytosis by melanocytes and internalization by keratinocytes, as well as how melanin is stored within these cells in a compartment that we propose be named the melanokerasome. Finally, we review the state of the art on the molecular mechanisms that lead to melanokerasome positioning above the nuclei of keratinocytes, forming supranuclear caps that shield the nuclear DNA from UV radiation. Thus, we provide a comprehensive overview of the current knowledge on the molecular mechanisms regulating skin pigmentation, from melanin exocytosis by melanocytes and internalization by keratinocytes to processing and polarization within keratinocytes. A better knowledge of these molecular mechanisms will clarify long-lasting questions in the field that are crucial for the understanding of skin pigmentation and can shed light on fundamental aspects of organelle biology. Ultimately, this knowledge can lead to novel therapeutic strategies to treat hypo- or hyper-pigmentation disorders, which have a high socio-economic burden on patients and healthcare systems worldwide, as well as cosmetic applications.

## 1. Introduction

The skin is the largest organ of the body and plays fundamental roles in water balance, regulation of body temperature, and protection against foreign agents and ultraviolet radiation (UVr) [1]. The upper layer of the skin—the epidermis—is mainly comprised of keratinocytes, which account for >80% of the cells in this layer. Keratinocytes stratify into five different *strata* according to their differentiation status: *stratum basale*, *stratum spinosum*, *stratum granulosum*, *stratum lucidum* (only visible in thick skin), and *stratum corneum* [2,3]. The epidermis also contains melanocytes, which are specialized cells derived from the neural crest during embryonic development [4]. Melanocytes are dispersed throughout the *stratum basale* and can extend dendrites to contact up to 40 keratinocytes, forming “epidermal-melanin units” (Figure 1A) [5,6]. The function of melanocytes is to synthesize the pigment melanin, subsequently transferring it to keratinocytes [7]. There are two types of melanin: black/brown eumelanin and yellow/red pheomelanin [8,9]. These are synthesized in melanocytes within melanosomes, which are lysosome-related organelles (LROs), as they are endosomal-derived, contain lysosomal markers, and are acidic in the early stages of their biogenesis [10]. Melanosomes display four distinct stages of maturation: In stages I and II, they are non-pigmented, initially resembling multivesicular endosomes. While pheomelanosomes maintain a spherical shape, eumelanosomes acquire an elliptic shape due to internal fibril nucleation of the premelanosomal protein (PMEL) [10]. Melanogenic enzymes required for eumelanin synthesis, including tyrosinase and tyrosinase-related proteins (TYRP1/2), are recruited to stage III melanosomes, and melanin starts to progressively deposit onto PMEL fibrils until melanosomes achieve full pigmentation (stage IV) [10,11]. Mature melanosomes accumulate at melanocyte dendrites and are then transferred to keratinocytes. After being internalized by keratinocytes, melanin is processed, i.e., trafficked, before accumulating in the supranuclear area, where it exerts its function of protecting the DNA of these cells from UVr [12]. Thus, skin pigmentation results from several processes that involve melanin synthesis in melanocytes, transfer from melanocytes to keratinocytes, and processing within keratinocytes. The molecular machinery involved in melanin synthesis and transport within melanocytes is now well characterized, but the mechanisms of melanin transfer remain controversial and poorly characterized. Moreover, only a few studies have addressed how melanin processing occurs within keratinocytes, and much remains to be uncovered. Here, we discuss the current knowledge and research priorities on the mechanisms of melanin transfer and processing. This knowledge is fundamental to understanding how skin pigmentation is sustained and can provide therapeutic strategies for pigmentary disorders, as well as cosmetic applications.

## 2. Melanin Transfer Models

Melanin must be transferred to keratinocytes to exert its photoprotective function in these cells. The process by which melanin transfer occurs has been a controversial topic since as far back as the 1960s, when melanin was first observed being transferred from a melanocyte to an epithelial cell (keratinocyte) [13]. As many as four models have been proposed to explain how melanin is transferred from melanocytes to keratinocytes, even though some could be variations of the same: (I) Cytophagocytosis of melanocyte dendrite tips, containing melanosomes, by keratinocytes; (II) Membrane fusion between keratinocytes and melanocytes and direct transfer of melanosomes; (III) Shedding of melanosome-laden globules by melanocytes and subsequent phagocytosis by keratinocytes; and (IV) Exocytosis of the melanin core by melanocytes, followed by phagocytosis by keratinocytes (Figure 2). These are subsequently summarized:Cytophagocytosis of melanocyte dendrites by keratinocytes.

This transfer mode was first proposed after observations made by electron microscopy (EM) of melanocyte dendrite tips within keratinocytes in vitro [14]. The process can be divided into four steps: Firstly, a melanocyte dendrite tip comes into contact with the keratinocyte plasma membrane and is subsequently engulfed by it; secondly, the melanocyte dendrite tip is pinched off, leading to the formation of a cytoplasmic vesicle filled with melanosomes; and thirdly, the compartment where melanin resides, which is theoretically limited by three membranes of different origins—the innermost one corresponding to the membrane of the melanosome and the plasma membranes of the melanocyte and the keratinocyte—fuses with the lysosome to form a phagolysosome. This leads to the degradation of the luminal membranes [15]. Lastly, the phagolysosome fragments into smaller vesicles containing aggregates or single melanin granules dispersed in the cytoplasm [16].
2.Fusion of melanocyte and keratinocyte membranes.

This model postulates that the plasma membranes of the melanocyte and keratinocyte fuse, allowing the formation of a conduit (e.g., filopodia or nanotubes) between the cytoplasm of the neighboring cells, through which melanosomes are transported [17]. As in the cytophagocytosis model, keratinocytes receive membrane-bound melanosomes. However, melanin within keratinocytes was shown by us and the group of Graça Raposo to be devoid of melanosome markers [18,19]. A possible explanation for this discrepancy is that the melanosome membrane is degraded inside keratinocytes by the endolysosomal system. It should be noted that this model and the cytophagocytosis one could represent variations of the same. Indeed, the group of Desmond Tobin has proposed a filopodia-cytophagocytosis model, in which melanocytes extend filopodia that bind to the keratinocyte plasma membrane and are then pulled and subsequently phagocytosed by the keratinocyte [20,21].
3.Shedding of melanosome-laden globules.

This model proposes that the blebbing of the melanocyte membrane leads to the formation and release of globules (1–3 µm in size) filled with melanosomes, which are then phagocytosed by keratinocytes [22,23]. Recently, other studies showed that globules filled with melanosomes are secreted from primary human melanocyte dendrites to the cell culture medium [24]. These melanosome-laden globules, when isolated from the medium and incubated with primary human keratinocytes, are internalized and processed in a similar manner as postulated by the cytophagocytosis model [25]. Indeed, according to this model, upon internalization, melanin is surrounded by three membranes. After processing, single membrane vesicles containing melanin are dispersed inside keratinocytes [25].
4.Coupled exocytosis/phagocytosis of the melanin core.

This transfer model predicts that melanosomes fuse with the melanocyte plasma membrane, leading to the exocytosis of the naked melanin core, termed the melanocore, to the extracellular space and its subsequent internalization by keratinocytes. Consequently, melanocores become surrounded by a single membrane derived from the keratinocyte plasma membrane upon internalization. The first evidence supporting this model came with the observation of extracellular melanin in human hair and skin, which was hypothesized to be internalized by keratinocytes in aggregates or individual granules [26]. More recently, we and others provided further evidence supporting this model, in which melanocores were observed in the interstitial space between melanocytes and keratinocytes [18,19]. We also showed that melanocores within keratinocytes are surrounded by a single membrane lacking the melanosome marker TYRP1, implying that the original melanosome membrane is lost during transfer [19,27].

The different models are supported by evidence obtained In various experimental systems, namely primary cells, cell lines, and tissue explants from different species, including mice, guinea pigs, chickens, frogs, and humans. In co-cultures of melanocytes and keratinocytes isolated from black guinea pig ears, evidence for the cytophagocytosis transfer model was reported [14]. In frog skin cells and tissues, melanin transfer was described as occurring through shed melanin-laden globules or exo-/phagocytosis of melanosomes, as extracellular melanin was found to be membrane-bound [23]. In chicken embryonic skin, melanin was proposed to be transferred via shed globules [28]. In mice, melanin transfer occurs in the hair follicles, and evidence for models of exo-/phagocytosis and shed globules has been described [29]. Regarding the different systems used, most studies were performed with cultured melanocytes, either alone or in co-culture with keratinocytes. However, in the skin epidermis, melanin transfer occurs between melanocytes and keratinocytes arranged in epidermal-melanin units. Therefore, models that reproduce this arrangement should be favored. Additionally, the use of melanoma cell lines for the study of the molecular mechanisms of pigment transfer should be performed with caution, as they can show alterations in melanin production, processing, and secretion [30]. Importantly, the model of exo-/phagocytosis is the one that collected more supporting evidence from the analysis of human skin sections and reconstructed human skin/epidermis in vitro models [19,31]. The shed globules is the other model that has been more explored. Furthermore, the four models can be essentially divided into the one that postulates the transfer of naked melanin (melanocore) and the ones that predict the transfer of membrane-bound melanosomes. It should be noted that it cannot be excluded that several mechanisms of melanin transfer co-exist, even in the same organism, as adaptations to different physiological or pathological conditions. To solve these long-lasting questions, a better understanding at the cellular and molecular levels of the pathways and regulators of melanin exocytosis from melanocytes and internalization by keratinocytes is essential.

## 3. Melanosome Transport and Exocytosis

Melanosome transport to the periphery of melanocytes precedes their transfer to neighboring keratinocytes [32,33,34]. Several regulators of melanosome transport on the cytoskeleton of melanocytes have been identified. The Ras-related protein in brain (Rab) 1, which mediates endoplasmic reticulum (ER)-Golgi trafficking, interacts with the SifA and kinesin-interacting protein (SKIP), as well as the anterograde microtubule-dependent motor kinesin 1, to regulate melanosome anterograde transport [35,36] (Figure 3A). Retrograde transport on microtubules is mediated by the intra-Golgi and endosomal trafficking regulator Rab36 and the product of the *dilute suppressor* gene, melanoregulin, which form a complex with Rab-interacting lysosomal protein (RILP) and the dynein-interactor dynactin subunit 1 (DCTN1) [37,38] (Figure 3A). Similarly, the secretory protein Rab44 was reported to form a tripartite complex with dynactin and the microtubule-dependent motor dynein to regulate melanosome retrograde transport [38,39] (Figure 3A). At the cell periphery, melanosomes are anchored to the cortical actin network and unidirectionally transported to the plasma membrane by the tripartite complex composed of the secretory protein Rab27a, the adaptor protein melanophilin, and the actin-dependent motor protein myosin Va [40,41] (Figure 3A). Despite the increasing knowledge about the molecular mechanisms that regulate melanosome positioning within melanocytes, it is not yet fully understood what their relative importance is and what the position itself means for the efficiency of melanin transfer.

Interestingly, another actin-dependent motor—myosin X—was shown to regulate melanin transfer to keratinocytes through filopodia/nanotube assembly [20]. Moreover, the N-methyl-D-aspartate receptor, the cell-cell adhesion molecule E-cadherin, and the recycling endosome protein Rab17 were also implicated in melanin transfer through filopodia formation [21,42,43]. Furthermore, the plasma membrane proteins caveolin-1 and cavin-1, which are required for plasma membrane invaginations called caveolae, were shown to be required for melanocyte-keratinocyte contacts, melanocyte dendricity, and melanin transfer [44]. However, it is not clear what transfer model(s) this process could involve.

Our group showed a key role for the recycling endosome protein Rab11b and two subunits of the exocyst tethering complex (Exo70 and Sec8) in melanin exocytosis and transfer, supporting the exo-/phagocytosis model [19,45] (Figure 3A). Recently, we showed that soluble factors derived from differentiated spinous/granular keratinocytes increase melanin exocytosis from melanocytes when compared with soluble factors derived from undifferentiated keratinocytes. We also found that the secretory Rab protein Rab3a enhances melanin exocytosis and transfer to keratinocytes under stimulation from differentiated spinous/granular keratinocytes [46] (Figure 3A). Therefore, these studies support the existence of a Rab11b-dependent basal route of melanin transfer and a Rab3a-dependent one, stimulated by keratinocytes. Importantly, we observed that Rab3a-stimulated melanin exocytosis is triggered by keratinocyte-derived soluble factors rather than extracellular vesicles [46]. Nevertheless, previous studies reported that keratinocyte-derived extracellular vesicles stimulate melanin synthesis by melanocytes [47]. Thus, we hypothesize that the Rab3a-stimulated melanin exocytosis pathway is required when melanin transfer needs to be upregulated, for instance, upon UVr exposure, also called facultative pigmentation. Additionally, Rab11a was found to regulate melanin exocytosis upon Toll-like receptor-2 (TLR-2) activation [48]. Finally, the actin remodeling small GTPase Ras homolog gene family A (RhoA) was shown to be required for membrane blebbing and release of melanosome-laden globules in chicken embryonic skins [28].

A better understanding of the putative feedback loops linking melanin synthesis, melanosome transport, and exocytosis will help clarify why different pathways can operate in different contexts to achieve the transfer of melanin. Moreover, it will be crucial to understand if the molecular regulators of melanin exocytosis can act together or in distinct pathways. Finally, future studies should explore if distinct stimuli require different proteins to regulate melanin exocytosis and if the different molecular regulators are related to different melanin transfer modes and pigmentation levels in human skin.

## 4. Melanin Internalization

Efficient skin pigmentation and photoprotection depend on the capacity of keratinocytes to internalize melanin. However, the mechanisms regulating melanin internalization by keratinocytes remain poorly understood. Recent studies using mouse and human cells reported that the internalization of secreted melanin (melanocores or melanosome-laden globules) by keratinocytes is likely to be the main mode of melanin transfer in the skin [49,50]. Given the physiological size of melanosomes (0.3–0.5 μm in diameter), only macropinocytosis and phagocytosis are predicted to allow the internalization of such large cargo [51,52,53,54]. Although keratinocytes are not professional phagocytes, they have been shown to efficiently phagocytose latex beads and bacteria [55,56]. Additionally, the stimulation of keratinocyte phagocytic capacity results in an increase in melanin transfer, strongly suggesting that phagocytosis is involved in melanin internalization by keratinocytes [50,56,57].

Both macropinocytosis and phagocytosis are highly dependent on actin cytoskeleton remodeling, which is regulated by Rho small GTPases [58]. Accordingly, melanin internalization was shown by us and others to require dynamic actin remodeling and depend on Rho GTPases [17,49,50]. Interestingly, melanocores and melanosomes require different Rho GTPases to be internalized: while melanocores are internalized in a Rac1-and Cdc42-dependent manner, melanosome internalization follows a RhoA-dependent route [49] (Figure 3B). These observations are in line with a previous study showing that the internalization of melanosome-rich globules upon TLR3 stimulation is dependent on the activity of RhoA and requires Cdc42 [50]. Furthermore, we observed that melanosome but not melanocore internalization is impaired by the inhibition of micropinocytosis [49]. Altogether, these observations suggest that different mechanisms are involved in melanin recognition and internalization by keratinocytes and that melanocores are internalized by phagocytosis and melanosomes by macropinocytosis. Importantly, there is no evidence for the transfer of single melanosomes, although it represents an excellent control since the only difference between melanocores and melanosomes is the presence of a surrounding membrane on melanosomes.

The specific recognition of melanocores by a membrane receptor is likely required for their phagocytosis by keratinocytes. The best known receptor involved in melanin uptake by keratinocytes is the protease-activated receptor-2 (PAR-2) [59]. In the epidermis, PAR-2 is highly expressed in the granular layer (*stratum granulosum*) and involved in epidermal barrier maintenance, keratinocyte differentiation, and pigmentation [60]. This seven-transmembrane G-protein-coupled receptor (GPCR) is activated by serine proteases (e.g., trypsin) that cleave a specific region of the PAR-2 extracellular N-terminal domain. Therefore, a new N-terminal sequence is exposed (SLIGKV [Ser-Leu-Ile-Gly-Lys-Val-NH_2_] in humans), which autoactivates the receptor by binding to a conserved region, leading to a change in receptor conformation and intracellular signaling [61,62]. PAR-2 activation using trypsin or synthetic SLIGKV enhances keratinocyte phagocytosis in vitro, while soybean trypsin inhibitors, which inhibit PAR-2, reduce it [56,59,63,64]. In vivo, the synthetic serine protease inhibitor RWJ-50353 inhibits PAR-2 activation, prevents UV-induced pigmentation, and enhances depigmentation in a concentration-dependent manner [65]. Interestingly, the levels of PAR-2 expression correlate with skin phototype, i.e., dark skins exhibit higher expression levels and activity throughout the epidermis when compared to light skins, where PAR-2 expression and activity are concentrated at the lower layers [66]. Importantly, in vitro and in vivo PAR-2 stimulation increases keratinocyte melanin content, and its activation was shown to enhance the keratinocyte phagocytic capacity of melanin, beads, and *Escherichia coli* K-12 [56,59,64,65,67]. Moreover, PAR-2 activation in keratinocytes signals through Rho GTPases [49,63], which are key players in the phagocytic process and are required for melanin internalization, as referred to. Furthermore, our group showed that melanocores but not melanosomes are dependent on PAR-2 for their uptake by keratinocytes, supporting the idea that melanocores are the form of melanin transferred to keratinocytes and the model of exo-/phagocytosis [27,49] (Figure 3B). Recently, it was also shown that the cation channel transient receptor potential ankyrin 1 (TRPA1) is required for the activity of PAR-2 in regulating melanocore phagocytosis by keratinocytes [68]. Apart from PAR-2, the stimulation of other receptors, including TLR-3 and fibroblast growth factor receptor 2/keratinocyte growth factor receptor (FGFR-2/KGFR), were described as being involved in melanin phagocytosis by keratinocytes [50,57,69] (Figure 3B). Interestingly, while our group showed evidence for the specific activation of PAR-2 by melanocores, others showed that TLR-3 can regulate melanocore and melanosome internalization by keratinocytes [27,49,50]. Therefore, these reports reinforce the notion that the way melanin is presented to keratinocytes, i.e., naked melanin in melanocores or membrane-bound melanosomes, results in the activation of different receptors and possibly distinct downstream signaling pathways [27,49,50]. Of note, FGFR-2/KGFR expression inversely correlates with skin color, i.e., it is more expressed in light skin keratinocytes. How to reconcile this with the pattern of PAR-2 expression, which is higher in dark skins, would be interesting to explore in future studies to better understand the contribution of different forms of internalized melanin (membrane-bound melanosome-laden globules or melanocores) to the skin phototype. Finally, the role of UVr in the stimulation of melanin internalization is also worth mentioning. PAR-2 expression and activity, as well as FGFR-2/KGFR activity, can be enhanced by exposure to UVr, increasing melanin transfer [70,71]. Moreover, a recent publication showed that the α7 nicotinic acetylcholine receptor (nAChR) responds to UVr exposure by increasing melanosome phagocytosis by keratinocytes [72].

Although phagocytosis has been widely assumed to be the internalization route followed by melanin, clear evidence for this has been lacking until recently. Since phagocytosis is a receptor-mediated process, it requires the specific recognition of the cargo to be internalized [73,74]. The evidence showing that melanocores and melanosomes require different regulators for their internalization [49] further supports the idea that melanin is recognized differently by keratinocytes depending on the presence or absence of a surrounding membrane. Whether melanosome-laden globules behave as melanosomes would be important to address in future studies. Additionally, the molecular determinants present on melanocores that are required for their specific recognition are not known and should be investigated to allow the identification of the phagocytic receptor involved. Importantly, beads, which have been widely used as a control for melanin internalization, most likely lack the molecular determinants required for the specific recognition of melanin. Therefore, despite the effort to use beads with sizes similar to melanosomes, they are likely not a suitable model to study melanin internalization by keratinocytes.

Even though there is abundant evidence for the important role of PAR-2 and other receptors (e.g., FGFR2/KGFR, TLR-3) in melanin internalization, their function in this process is likely to be indirect, i.e., increasing the overall keratinocyte phagocytic efficiency, rather than being directly involved in melanin recognition [50,56,57,69]. In fact, GPCRs like PAR-2 were shown to recognize chemoattractants secreted by microorganisms, enhancing the phagocytic capacity through the remodeling of the actin cytoskeleton [75,76,77]. Thus, PAR-2 can be required for chemotaxis, i.e., directing keratinocytes towards melanin and facilitating its phagocytosis. Similarly, although TLRs can recognize molecular patterns on different cargoes, they are not phagocytic receptors, cooperating instead to prime phagocytosis [78,79]. Additionally, PAR-2 is highly expressed in the granular layer [60], whereas melanin transfer mostly occurs in the basal layer [80,81,82], which suggests a paracrine regulation of melanin internalization by suprabasal keratinocytes. This further implies that PAR-2 is likely not involved directly in melanin recognition but instead in the regulation of keratinocyte phagocytic activity. Thus, how melanocores are recognized by keratinocytes to trigger their internalization through phagocytosis remains to be addressed. Among the possible candidates are the C-type lectin receptors, especially considering the recent study showing that MelLec receptors, expressed in endothelial and myeloid cells, specifically recognize fungal melanin [83]. Interestingly, keratinocytes express the C-type lectin receptor Dectin-1, which is known to be involved in wound re-epithelialization and keratinocyte-mediated innate immune responses [84,85]. Despite the differences between fungal and mammalian melanin, the involvement of C-type lectin receptors such as Dectin-1 in human melanin recognition is worth testing. Remarkably, keratinocytes express Fc-γ receptors, as described in the context of inflammatory skin diseases such as psoriasis [86,87]. To the best of our knowledge, the involvement of Fc-γ receptors in melanin internalization by keratinocytes has not been studied so far but might be interesting to explore in future studies. Fc-γ receptors require the binding of an opsonin (e.g., an antibody) to the particle internalized. This raises the question of whether melanin also requires an opsonin to be recognized and internalized by keratinocytes.

The identification of the phagocytic receptor(s) involved in melanin internalization can provide an appealing target to modulate this process and, consequently, skin color. Furthermore, the characterization of the receptor(s) and downstream signaling pathways involved in melanin internalization can also shed light on the mechanism regulating melanin processing and positioning within keratinocytes, as discussed below.

## 5. Melanin Processing

Melanin processing refers to the steps that occur within keratinocytes after melanin internalization. Remarkably, the mechanisms involved remain elusive, despite keratinocytes being the melanin recipient cells, melanin exerting a photoprotective effect, and differences in melanin processing linked to skin phototype. Indeed, while in light skin melanin mainly accumulates within keratinocytes of the basal layer of the epidermis, in dark skin melanin is also present in keratinocytes of the suprabasal layers [80,81,82] (Figure 1A). Importantly, it is well established that melanin organization within keratinocytes differs between phototypes [18,88,89,90,91]. In light skins, melanin is present in clusters, whereas dark skins predominantly contain single melanin granules that are usually larger, even though melanin-containing organelles display a similar size in light and dark skins [18,88,89,90,91] (Figure 1B). The mechanisms involved in the establishment of clusters or single granules are essentially unknown, except that they depend on the skin phototype of the keratinocyte and are independent of the type of melanin transferred (eumelanin vs. pheomelanin) [92].

Melanin turnover in melanocytes and keratinocytes has been suggested to be controlled by autophagy [93,94,95]. In keratinocytes, the autophagic/lysosomal activities were described to inversely correlate with skin color, i.e., higher in keratinocytes derived from light skins than in those from dark skins, independently of the keratinocyte differentiation status [93,96,97,98,99,100] (Figure 4B). Moreover, the pharmacological or genetical modulation of autophagy in distinct experimental conditions (2D or 3D cultures and skin explants) was shown to alter keratinocyte melanin content [93,95,101,102,103] (see Table 1). Indeed, autophagy induction was found to lead to a reduction in the levels of melanin within keratinocytes, whereas autophagy inhibitors lead to an increase in these levels. This was attributed to melanin degradation vs. persistence, respectively [93,95,101,102,103]. Remarkably, the decline in autophagic activity in the epidermis with aging results in the accumulation of melanin, namely within keratinocytes, forming hyperpigmented lesions such as those seen in senile lentigo [101]. Interestingly, autophagy induction in aged skin was suggested to restore pigmentation, reinforcing the importance of autophagy in melanin turnover [101].

Notwithstanding these observations, melanin is not found in double membrane organelles, i.e., autophagosomes, within keratinocytes, and melanin-containing compartments lack the classical autophagosomal protein LC3 [18,104]. Indeed, melanin is found in single membrane compartments inside keratinocytes, supporting exo-/phagocytosis or the fusion of melanocyte and keratinocyte membrane transfer models [17,18,19] (Figure 4A). Despite the lack of clear evidence, it has been proposed that in the cytophagocytosis and shed globules models, the melanin surrounding membranes are degraded during processing within keratinocytes [18,52,105]. For instance, it was suggested that during melanosome internalization, the melanosome membrane protein TYRP1 is quickly removed, possibly through degradation, but the mechanisms involved were not investigated [106]. This results in the formation of single membrane melanin-containing organelles within keratinocytes. The characterization of the molecular processes involved in the formation of these organelles is important to better understand the fate of the pigment in keratinocytes, and only recently has this question started to be addressed. Graça Raposo’s group showed that, independently of the skin phototype, the lysosomal-associated membrane protein 1 (LAMP1) and the tetraspanin cell surface antigen 63 (CD63)—markers of late endosomes (LEs)/lysosomes—are present in the membrane of single and clustered melanocores [18] (Figure 4A). Additionally, we found that in mouse keratinocytes, melanin-containing compartments are positive for the early endocytic markers early endosome antigen 1 (EEA1) and Rab5, as well as for LAMP2 and CD63 [27] (Figure 4A). Noteworthy, these experiments were performed at a fixed timepoint, and, therefore, the maturation and dynamics of the compartment were not assessed. Accordingly, several Rab small GTPases, including the LE/lysosomal Rab7b, were found to localize to melanin-containing compartments inside mouse keratinocytes [106].

Moreover, we and Graça Raposo’s group showed that melanin accumulates in moderately acidic compartments (presenting low levels of Lysotracker and DAMP [3-(2,4-dinitroanilino)-3′-amino-N-methyldipropylamine] accumulation) with low degradative capacity, as shown by the low levels of dye quenched-bovine serum albumin (DQ-BSA) fluorescence, despite the abundance of these types of organelles within keratinocytes [18,27] (Figure 4A). Indeed, we reported that in cultured mouse keratinocytes, melanin is long-lived and persists for long periods [27], which suggests that melanin-containing compartments are specialized for long-term melanin storage inside keratinocytes. We termed these melanin-containing compartments inside keratinocytes melanokerasomes (MKSs) [52] to distinguish them from melanosomes in melanocytes. MKSs are likely unique melanin-storage compartments presenting low acidity and degradative capacity but maintaining characteristics of LEs/lysosomes, including the membrane markers LAMP1/2 and CD63 [18,27] (Figure 4A). Given that CD63 is frequently associated with LROs in specialized cell types and that LROs also present moderate luminal acidity and degradative capacity when compared to lysosomes, it has been suggested that MKSs could be a new type of LRO [18,107]. Although this is an interesting hypothesis that gives MKSs additional relevance as functional organelles, MKSs start as phagosomes and are thus not endosomal-derived or known to receive cargo from the Golgi. Additionally, there is no evidence that MKSs can undergo regulated exocytosis, the most notable distinguishing feature of LROs. Another possibility is that MKSs are lysosome-derived organelles or represent a dysfunctional lysosome that has lost its degradative ability. Indeed, this could explain melanin persistence inside keratinocytes. However, it cannot be discarded that melanin is resistant to degradation (see below). Thus, future studies should characterize MKS formation in a dynamic way, i.e., considering its spatiotemporal maturation and interaction with other organelles, as well as how they enable melanin persistence. This will help understand the biogenesis and lifecycle of these organelles specialized for melanin storage.

Another relevant question is whether melanin internalization is linked to its fate within keratinocytes. This applies not only to the influence of the transfer mode on the way melanin is processed (the transfer of melanosomes, melanosome-laden globules, or melanocores), but also to the contribution of the signaling pathways activated downstream of membrane receptors such as PAR-2, FGFR-2/KGFR, and TLR-3 (Figure 4A). For instance, PAR-2 was shown to decrease autophagy through the phosphatidylinositol 3-kinase (PI3K)/protein kinase B (Akt)/mammalian target of rapamycin (mTOR) pathway in kidney cells [108,109]. Moreover, melanin degradation was found to be promoted when this pathway is inhibited using the dietary flavonoid isoliquiritigenin and, consequently, autophagy is activated [102] (Table 1). Indeed, the contribution of different melanin internalization routes to processing is unknown [51]. Interestingly, previous work from our lab showed that melanin clusters can be internalized by keratinocytes in the skin [19]. This raises several questions, including whether the clusters found in light skins are formed during internalization or after processing inside keratinocytes. It is also unclear whether melanin processing is different among phototypes and how this contributes to the distinct organization patterns observed in light and dark skins. In particular, the increased autophagic/lysosomal activities in light skins, when compared with dark ones, suggest the existence of distinct melanin degradation capacities by keratinocytes from different phototypes [93,96,97,98,99,100] (Figure 4B). This might also explain the detection of suprabasal melanin in dark skins, possibly resulting from decreased or delayed melanin degradation (Figure 4B). Thus, it is crucial to clarify the contribution of melanin internalization and autophagy to melanin organization within keratinocytes. It is likely that MKS formation and processing, starting with melanin internalization and culminating in the establishment of a unique storage compartment, the MKS, are determined by several factors.

Strikingly, there is contradictory evidence regarding melanin degradation within keratinocytes, and it is not clear if melanin can be degraded at all. Given its protective properties, melanin should be preserved, and the characteristics of the MKS, namely moderate acidity and degradative capacity, point towards that [18,27]. In fact, melanin solubility increases in alkaline conditions, whereas it remains insoluble and precipitates in an acidic milieu [110,111]. Moreover, attempts to destroy melanin hydrolytically in vitro result in protein and lipid degradation but preservation of the melanin moiety [110,111]. Therefore, whether the formation of MKSs is required for melanin preservation and/or degradation within keratinocytes remains to be established. Interestingly, fungal melanin was shown to inhibit LC3-associated phagocytosis (LAP) and promote *Aspergillus fumigatus* pathogenicity [112,113]. Moreover, we proposed that melanin shares with microorganisms the capacity to subvert intracellular trafficking pathways to resist degradation [114]. Hence, the potential role of LAP in melanin processing should be tested. Although melanin is long-lived inside keratinocytes in vitro, it localizes mostly to the basal layer, and its accumulation throughout the epidermis is only evident in darker phototypes, where suprabasal keratinocytes retain degraded melanin fragments known as “melanin dust”. Of note, the detection of melanin in the corneal layer ( “melanin dust”), observed when using Fontana Masson staining, can hardly be found by EM and is therefore attributed to an artifact of this staining [27,82,92,115,116]. This raises once again the question of whether melanin can be degraded during keratinocyte differentiation and if the right tools are available to perform this analysis accurately [116]. Recently, melanin accumulation in the basal layer has been attributed to the asymmetric division of keratinocytes, with basal keratinocytes retaining the pigment [104]. Importantly, the same study suggested that in stress conditions (e.g., regenerating skin), the symmetric division is prioritized, leading to a broader melanin distribution between the daughter and progenitor keratinocytes [104]. Therefore, if melanin concentration in the basal layer results from the absence of degradation, asymmetric cell division, or both, it should be evaluated. It should also be noted that melanin processing was studied using different models, including human skin explants, primary human keratinocytes, and mouse/human keratinocyte cell lines. Thus, despite the relevant observations made, it is urgent to develop new strategies to study the dynamics of melanin processing in more physiologically relevant models. Moreover, only the combination of different approaches will allow a reliable analysis of the fate of melanin inside keratinocytes.

## 6. Melanin Polarization

Melanin processing ends with its polarization to form supranuclear caps, or “parasols”, that protect the genetic material present in the nuclei of keratinocytes against the genotoxic effects caused by UVr exposure [117] (Figure 5). Indeed, previous studies showed a significant decrease in the formation of both cyclobutane pyrimidine dimers (CPDs) and (6–4) photoproducts (6–4PP) in cells with supranuclear melanin caps compared with cells without these [118]. Although the importance of the supranuclear melanin caps in protecting against UVr has been clearly established, the mechanisms behind melanin polarization in the skin remain to be uncovered. The formation of supra-nuclear melanin caps has been suggested to be stimulated by UVr [119]. Indeed, it was recently reported that UVA radiation induces melanin polarization through the G-protein-coupled receptor opsin 3 [120]. Nevertheless, our results suggest that supra-nuclear melanin caps can form in reconstructed human epidermis without UVr exposure [31]. Therefore, future studies should determine what drives melanin polarization and supra-nuclear cap formation. Interestingly, we have preliminary unpublished data suggesting that keratinocyte stratification, which involves proliferation, differentiation, and migration, can lead to melanin organization in supranuclear caps, akin to organelle polarization towards the leading edge of migrating cells [121].

Regarding the molecular players involved in melanin polarization, prior studies have shown that the intermediate chain of cytoplasmic dynein co-localizes with melanin both in cultured human keratinocytes and human skin explants [12]. The same study also reported that treatment with anti-sense DNA for cytoplasmic dynein heavy chain leads to a significant dispersion of perinuclear melanin in human keratinocytes. Subsequent studies demonstrated that the silencing of the dynactin p150^Glued^ subunit impairs the perinuclear clustering of microspheres in human keratinocytes [122], which suggests that melanin polarization is dependent on the cytoplasmatic dynein-dynactin motor complex (Figure 5). These results were further corroborated by recent studies, which reported a significant impairment in melanin polarization upon disruption of the actin cytoskeleton and the microtubule network [123]. The same study also reported the co-localization between melanin, the centrosome, and nuclear satellites at the apical region of stratified keratinocytes, suggesting a potential role for centrosome-related proteins in the regulation of melanin polarization. Thus, very little is still known about the molecular machinery involved in melanin polarization within keratinocytes. In particular, how retrograde dynein/dynactin-dependent transport is regulated and if the same machinery used by LEs/lysosomes is also employed by MKSs, remains to be determined. Given the association of Rab7b with MKSs [106] and the role that Rab7 plays in retrograde transport of LEs/lysosomes [124], it should be confirmed if Rab7 is also involved in melanin polarization.

## 7. Conclusions and Future Perspectives

In contrast with the mechanisms of melanin synthesis, melanosome biogenesis, maturation, and transport, those of melanin exocytosis, transfer, and processing are much less understood. Indeed, one of the topics most debated in the field remains that of melanin transfer between melanocytes and keratinocytes. Thus far, this issue has been studied mainly in cultured primary cells and cell lines, including from melanoma and diverse animal models. However, these present clear limitations for the study of pigmentation, as they cannot fully recapitulate the complexity of the human skin architecture and crosstalk that occurs in vivo. Furthermore, they can present changes in the mechanisms involved due to their transformation. Tissue explants have also been used, but despite recapitulating the physiology of skin pigmentation, they suffer changes when in culture. The advent of reconstructed pigmented skin/epidermis and, more recently, skin-on-a-chip models [125] has brought the possibility of manipulating melanocytes and/or keratinocytes and understanding the mechanism of skin pigmentation in a setting very close to the physiological one. Therefore, these models should be further explored to determine if distinct melanin transfer modes can occur in constitutive and facultative pigmentation, as well as in pathological conditions.

Regarding melanin exocytosis from melanocytes, several molecules have now been identified to regulate this process. However, we have just started to unveil the molecular mechanisms involved in basal and stimulated pathways. Thus, the receptors, signals, and molecules that can trigger stimulated melanin release should be addressed in future studies. Moreover, whether these signaling pathways and molecular regulators are the same or not under physiological (basal or facultative pigmentation) or pathological conditions (pigmentary disorders) remains to be determined. Undoubtedly, understanding better melanocyte-keratinocyte crosstalk mediated by both soluble factors, extracellular vesicles, and cell-cell contacts will be a key to solving this riddle. Indeed, melanocores were found very close to melanocyte and keratinocyte membranes in human skin explants [19,31], indicative of a tight connection between these two cell types, albeit without membrane fusion. This architecture resembles the well-known immunological and neuronal synapses [126,127], suggesting the existence of a pigmentary synapse through which keratinocytes and melanocytes can communicate to control the production and transfer of melanin in the epidermis. More studies are needed to characterize this putative pigmentary synapse.

There is now evidence that melanocores are internalized by keratinocytes through phagocytosis, but the phagocytic receptor is unknown. Therefore, the identification of the molecular determinants present in melanocores and the receptor(s) that specifically recognize them will be crucial to clarifying the transfer pathway(s) at the molecular level. Another fundamental question that remains to be addressed is whether the internalization route and downstream signaling pathways affect the fate of melanin regarding its processing within keratinocytes. The mechanism(s) behind the differences in melanin organization inside keratinocytes from distinct phototypes is also an intriguing question that remains to be solved [18,88,89,90,91]. Along with this, understanding how MKSs, which are unique melanin-storage organelles, are established and their further characterization will be essential to unveiling the mechanisms involved in fine-tuning melanin persistence within keratinocytes. Moreover, this can provide new insights into the adaptation of intracellular organelles to specialized functions. Unsolved questions also include the mechanisms through which autophagy regulates melanin turnover within keratinocytes, despite several studies showing that the modulation of this pathway alters the melanin content of keratinocytes [93,94,95]. Importantly, it must be determined if melanin can be degraded at all and if the turnover involves changes in melanin organization, i.e., single melanin granules in dark phototypes or melanin clusters more characteristic of light phototypes. These studies will have a great impact on the characterization of melanin dispersion throughout the epidermis in different phototypes, as well as the accumulation of melanin within keratinocytes in conditions such as post-inflammatory hyperpigmentation and senile lentigo [128]. Finally, it should be determined what the cues are for the polarization of melanin in supranuclear caps and how these are maintained throughout the life span of keratinocytes. The answers to these questions will shed light on fundamental mechanisms that are essential for skin pigmentation but have remained elusive until now and will provide novel targets to modulate skin pigmentation to treat pigmentary disorders, as well as for cosmetic purposes.

## Figures and Tables

**Figure 1 ijms-24-11289-f001:**
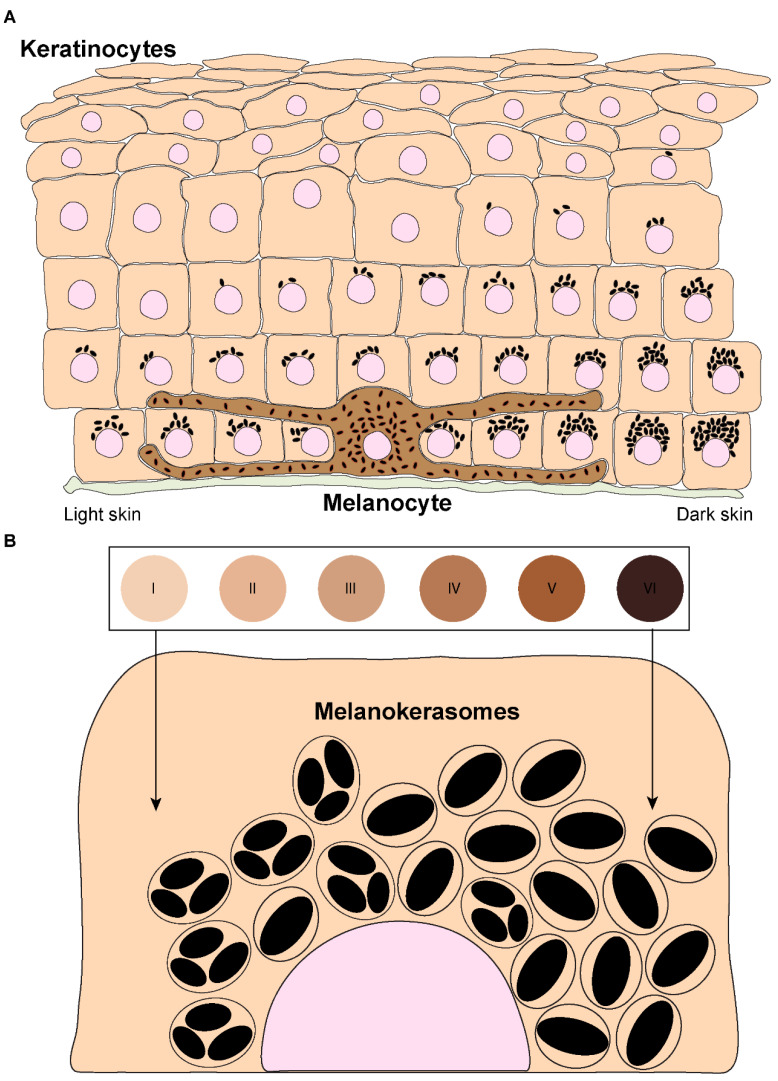
**Melanin distribution throughout the epidermis and organization within keratinocytes from distinct phototypes.** (**A**) Melanin dispersion in the epidermis is achieved through the formation of epidermal-melanin units. The accumulation of melanin is higher in the basal layers, regardless of the skin phototype (represented by the numbered circles and labeled from I—lower/lighter phototype—to VI—higher/darker phototype). However, in lower phototypes (lighter skins), less melanin is concentrated in the basal layer, whereas in higher phototypes (darker skins), higher levels can be found in the basal layer, and melanin is also present in the immediate suprabasal layers. (**B**) Melanin organizes differently within keratinocytes derived from different phototypes. In lower phototypes, melanin is mainly stored in clusters of several melanin granules within a single membrane organelle, while higher phototypes prominently accumulate single granules in single membrane organelles.

**Figure 2 ijms-24-11289-f002:**
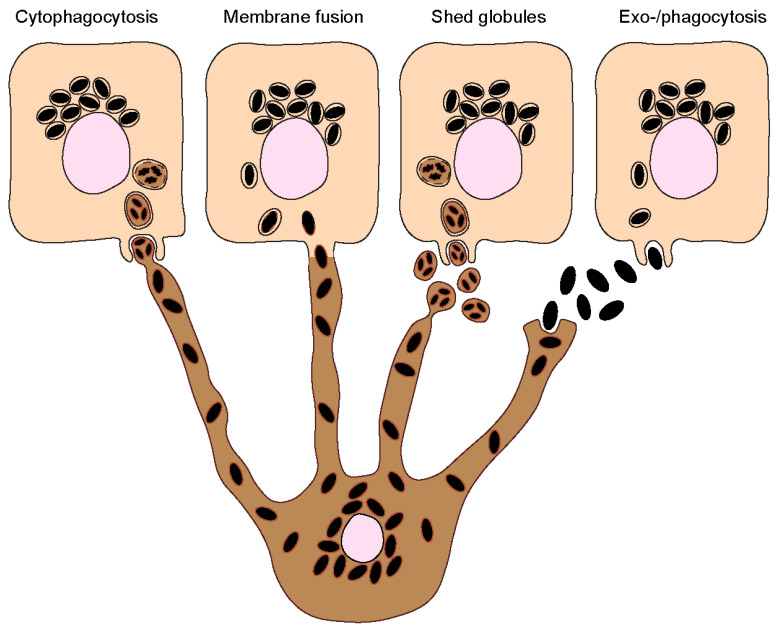
**Melanin transfer from melanocytes to keratinocytes.** Four different models have been proposed to explain the mechanism(s) of melanin transfer from melanocytes (dark-colored) to keratinocytes (light-colored), including the cytophagocytosis of melanocyte dendrites by keratinocytes; the direct fusion of melanocyte and keratinocyte membranes; the shedding of melanosome-laden globules by melanocytes; and the coupled exocytosis/phagocytosis of the melanin core. Despite the evidence published using different models, recent studies using human and mouse cell lines, as well as more complex models such as reconstructed human skins/epidermis and skin biopsies, have supported the transfer of melanin in globules or as melanocores.

**Figure 3 ijms-24-11289-f003:**
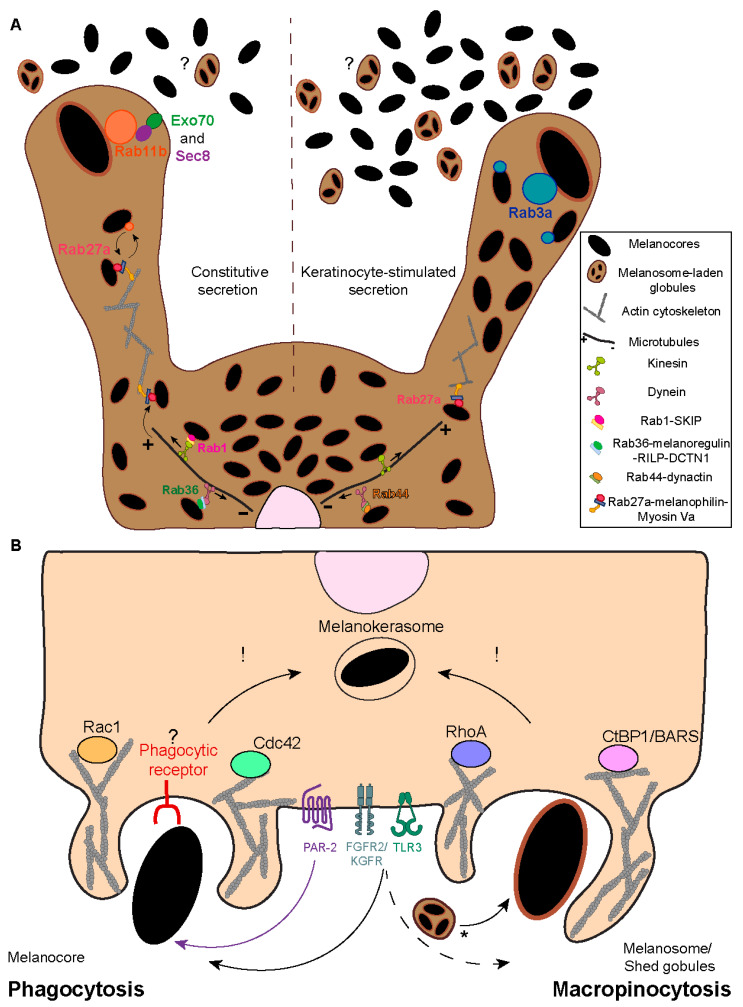
**Melanin exocytosis by melanocytes and internalization by keratinocytes.** (**A**) Melanin exocytosis in physiological conditions is thought to occur at the tips of melanocyte dendrites, requiring melanosome transport to the cell periphery. Anterograde long-range transport occurs via microtubules and is mediated by the complex Rab1-SKIP-kinesin 1. On the other hand, retrograde transport is regulated by Rab36-melanoregulin-RILP-DCTN1 and/or Rab44-dynactin-dynein. Melanosome positioning results from the balance between anterograde and retrograde transport. Since retrograde transport overrides anterograde transport, melanosomes tend to accumulate in the perinuclear region. Melanosome peripheral positioning is dependent on the tripartite complex composed of Rab27a-melanophilin-myosin Va. The small GTPase Rab11b and the exocyst complex subunits Exo70 and Sec8 were found to regulate basal melanin exocytosis and transfer. Furthermore, Rab3a enhances melanin exocytosis and transfer to keratinocytes under stimulation by soluble factors derived from differentiated keratinocytes. Whether melanosome-laden globules are also secreted from melanocytes using the same molecular regulators is not known (marked by “?”). (**B**) Recent studies found key differences in the molecular players involved in melanin recognition and internalization by keratinocytes according to the way melanin is presented to keratinocytes, i.e., “naked” melanocores or membrane-bound melanosomes. Melanocores were shown to be phagocytosed in a PAR-2-dependent manner by keratinocytes, requiring Rac1 and Cdc42 for efficient internalization. The existence and identity of the phagocytic receptor that recognizes melanocores remains elusive (marked by “?”) In contrast, purified membrane-bound melanosomes are internalized through macropinocytosis in a PAR-2-independent manner, requiring RhoA and CtBP1/BARS. Whether the internalization of melanosome-laden globules requires the same regulators as melanosomes remains to be determined (marked by “*”). Additionally, FGFR2b/KGFR and TLR3 stimulation were shown to enhance the internalization of melanosomes and melanocores, although the mechanisms were not characterized. Whether the existence of different internalization routes affects melanokerasome processing is not known (marked by “!”).

**Figure 4 ijms-24-11289-f004:**
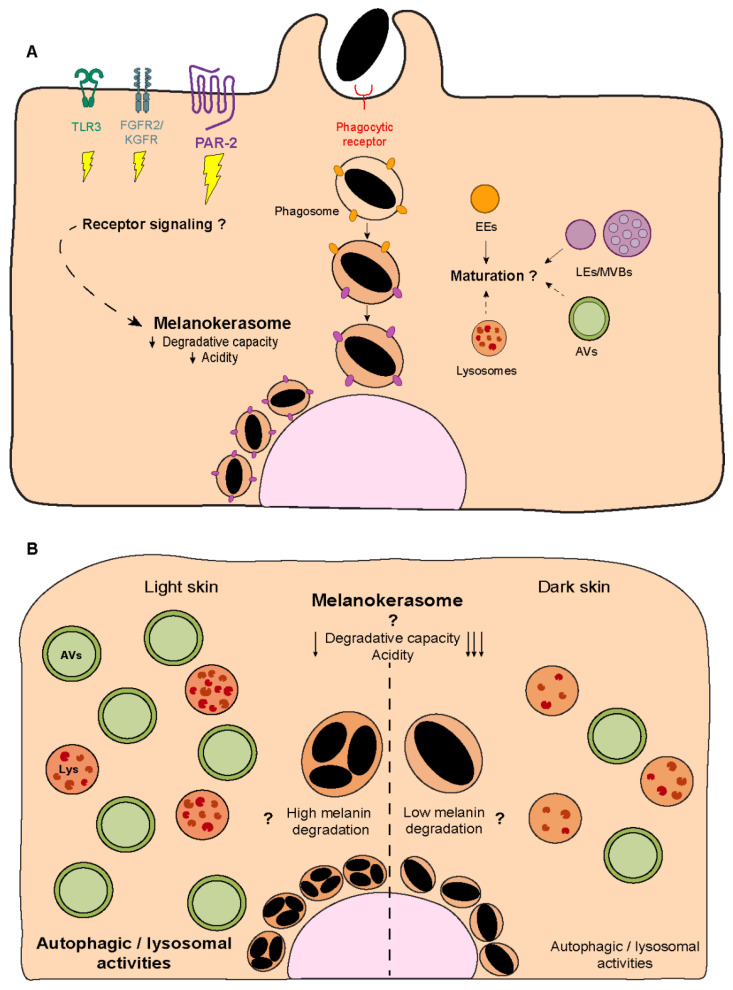
**Melanin processing within keratinocytes.** (**A**) Recent studies have proposed that transferred melanin inside keratinocytes is surrounded by early and late endocytic markers. Whether this organelle undergoes further maturation is not known, but our recent observations suggest that melanin is stored in a specialized organelle that we named melanokerasome (MKS), which has a poorly acidic lumen and non-degradative properties. The contribution of other organelles to the biogenesis of MKSs is not fully understood, despite the possible involvement of early endocytic, late endocytic, and autophagic vesicles. Additionally, the relevance of receptor signaling (e.g., PAR-2, KGFR/FGFR2, and TLR3 or phagocytic receptors) for the trafficking and biogenesis of MKSs is not known. (**B**) Autophagy has been reported to regulate melanin degradation within keratinocytes, although the mechanisms are not understood. Moreover, autophagic and lysosomal activities are higher in lighter skin when compared to darker skin. Nevertheless, it is not fully understood whether MKSs from distinct phototypes present different properties regarding their ability to allow melanin persistence or degradation (marked by arrows–more arrows mean lower degradative capacity and acidity), nor the contribution of autophagic/lysosomal activities (marked by “?”) for this. EEs—Early endosomes; LEs—Late endosomes; MVBs—Multivesicular bodies; AVs—Autophagic vesicles; Lys—lysosomes.

**Figure 5 ijms-24-11289-f005:**
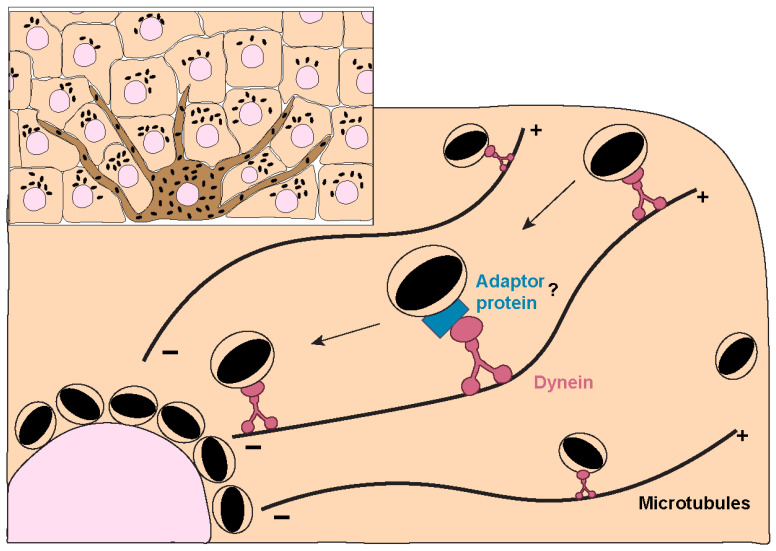
**Melanin trafficking and positioning within keratinocytes.** The accumulation of melanin in the supranuclear region of keratinocytes is crucial for its photoprotective properties in the skin. Despite the importance of this process, the molecular mechanisms involved are still poorly understood. Melanin is known to be retrogradely transported along microtubules in a process mediated by the dynein-dynactin motor complex. It is not known if an adaptor protein (or proteins, marked by “?”) is involved in this process or how the transport is regulated.

**Table 1 ijms-24-11289-t001:** List of compounds that increase or decrease autophagy in keratinocytes and lead to decreased or increased melanin content, respectively.

Autophagy Modulator	Mechanism Proposed	Model	References
Isoliquiritigenin	Decrease in melanin content through the suppression of the PI3K/AKT/mTOR pathway	Human epidermal keratinocytes	[102]
ATG7 silencing	Increase in melanin content	HaCaT human keratinocytes	[102]
3-Methyladenine (3-MA)	Increase in melanin content	HaCaT human keratinocytes	[102]
Pentasodium tetracarboxymethyl palmitoyl dipeptide-12 (PTPD-12)	Decrease in melanin content	Human epidermal keratinocytes	[95]
ATG7 silencing	Increase in melanin content	Human epidermal keratinocytes	[93]
ATG13 or UVRAG silencing	Increase in melanin content	Human epidermal keratinocytes	[93]
Lysosome inhibitors (E-64-D and pepstatin A)	Increase in melanin content	RHPE and skin explants	[93]
Verapamil	Decrease in melanin content	RHPE and skin explants	[93]
Rapamycin	Decrease in melanin content	RHPE and skin explants	[93]
Hydroxychloroquine	Increase in melanin content	RHPE and skin explants	[93]
Marliolide derivative (5-methyl-3-tetradecylidene-dihydro-furan-2-one; DMF02)	Decrease in melanin content through Nrf2-p62 activation	HaCaT and human epidermal keratinocytes	[103]
Torin 1	Decrease in melanin content through mTOR inhibition	Skin explants	[101]

ATG—autophagy-related protein; UVRAG—UV radiation resistance associated; RHPE—reconstructed human pigmented epidermis.

## Data Availability

Data sharing not applicable.

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
