# Peer review of "Melanin’s Journey from Melanocytes to Keratinocytes: Uncovering the Molecular Mechanisms of Melanin Transfer and Processing"

_ijms, 2023, doi:10.3390/ijms241411289_

Round 1
Reviewer 1 Report
Bento-Lopes et al described the molecular mechanisms of melanin transfer and processing in melanocytes and keratinocytes.
The authors addressed comprehensive literature review including recent research, which is educational for the readers.
Author Response
We thank the Reviewer for critically assessing our manuscript entitled “Melanin Journey from Melanocytes to Keratinocytes: Uncovering the Molecular Mechanisms of Melanin Transfer and Processing” and for the positive comments.
Reviewer 2 Report
This is a comprehensive and in-depth literature review covering the current research in the melanin transfer field. Despite the great unknowns and discrepancies on many mechanisms involved, the authors addressed the major mechanisms using great model figures and it is in general very clear and easy to understand. The manuscript is acceptable with a few minor comments.
1. A few places need to add citation from the introduction part. For example, Line 49-51 "These are synthesized in melanocytes...of their biogenesis."
2. The figure legends need to be more concise. Then current ones has redundant information that has been addressed in the main text.
3. It is unclear what the three "?"s in Figure 3A stands for, could the authors elaborate in the figure legends?
4. The right-half of Figure 4A is too busy and may cause confusion. Please re-organize this figure.
5. The authors mostly discussed the melanin transfer in human, but also mentioned studies in other species such as mice. Can the authors discuss on the difference in melanin transfer among species and/or systems?
Author Response
We thank the Reviewer for critically assessing our manuscript entitled “Melanin Journey from Melanocytes to Keratinocytes: Uncovering the Molecular Mechanisms of Melanin Transfer and Processing” and for the positive comments. We introduced the suggested changes in the manuscript (highlighted in yellow) and hope they address all the concerns raised.
Below, we include a detailed point-by-point response to the comments made.
- A few places need to add citation from the introduction part. For example, Line 49-51 "These are synthesized in melanocytes...of their biogenesis."
As requested, we added several new references to the introduction (Refs 3, 4, 10, 11, 12). Moreover, we corrected a mistake in the percentage of epidermal cells that are keratinocytes (also highlighted in yellow).
- The figure legends need to be more concise. Then current ones has redundant information that has been addressed in the main text.
We thank the Reviewer for this comment and simplified the figure legends avoiding redundant information already addressed in the text.
- It is unclear what the three “?”s in Figure 3A stands for, could the authors elaborate in the figure legends?
We added detailed information about the symbols in Figures 3A and 3B. To make it clearer, we changed some of the symbols (added “!” and “*” to Figure 3B).
- The right-half of Figure 4A is too busy and may cause confusion. Please re-organize this figure.
We thank the Reviewer for this comment and simplified Figure 4A. We hope it is easier to understand now.
- The authors mostly discussed the melanin transfer in human, but also mentioned studies in other species such as mice. Can the authors discuss on the difference in melanin transfer among species and/or systems?
As requested, we added additional information on the differences in melanin transfer among species and systems (lines 165-178). We thank the Reviewer for this suggestion on a topic that we consider to be very important for the field.
Reviewer 3 Report
This is an interesting and well designed review which clearly demonstrate the complexity of the melanocytes keratinocytes interaction in the pigmentary process and particularly in the melanosome transfer. The different processes are appropriately discussed. A few note could be added regarding the molecules used to control the pigmentation and which role they can have as for examples the PAR2 inhibitors
Author Response
We thank the Reviewer for critically assessing our manuscript entitled “Melanin Journey from Melanocytes to Keratinocytes: Uncovering the Molecular Mechanisms of Melanin Transfer and Processing” and for the positive comments. As suggested, we introduced a small note on the effect of PAR-2 modulation (i.e., stimulation or inhibition) on skin pigmentation levels (highlighted in yellow, lines 279-287). We hope this addresses the Reviewer’s suggestion.